# Non-destructive erosive wear monitoring of multi-layer coatings using AI-enabled differential split ring resonator based system

Vishal Balasubramanian[1], Omid Niksan[1], Mandeep C. Jain ®[1], Kevin Golovin ®[2] & Mohammad H. Zarifi ®[1] ✉

Unprotected surfaces where a coating has been removed due to erosive wear can catastrophically fail from corrosion, mechanical impingement, or chemical degradation, leading to major safety hazards, financial losses, and even fatalities. As a preventive measure, industries including aviation, marine and renewable energy are actively seeking solutions for the real-time and autonomous monitoring of coating health. This work presents a real-time, non-destructive inspection system for the erosive wear detection of coatings, by leveraging artificial intelligence enabled microwave differential split ring resonator sensors, integrated to a smart, embedded monitoring circuitry. The differential microwave system detects the erosion of coatings through the variations of resonant characteristics of the split ring resonators, located underneath the coating layer while compensating for the external noises. The system's response and performance are validated through erosive wear tests on single- and multi-layer polymeric coatings up to a thickness of 2.5 mm. The system is capable of distinguishing which layer is being eroded (for multi-layer coatings) and estimating the wear depth and rate through its integration with a recurrent neural network-based predictive analytics model. The synergistic combination of artificial intelligence enabled microwave resonators and a smart monitoring system further demonstrates its practicality for real-world coating erosion applications.

Erosive wear is a detrimental process that causes irreversible damage to the exterior surfaces of structures such as bridges, aircraft, automobiles, and naval infrastructure[1]. Erosion has been identified as the primary reason for many accidents, including the Genoa bridge collapse (Italy, 2018), the Bhopal gas tragedy (India, 1984), and the Carlsbad gas pipeline fire (USA, 2000), which have led to the loss of thousands of lives[2–4]. Financially, the direct cost due to the corrosion of metals strains the economy of the USA by about $276 billion per year, representing 3.1% of the USA gross domestic product[5]. Erosive wear is primarily caused due to friction, impact of debris, high wind speed, or liquid flow, and results in structural instability, safety hazards, and reduced lifetime[6]. As a low-cost solution, wear-resistant coatings (WRC) are commonly employed to protect the structure from erosive-wear. However, these coatings too have a limited service life and wear out with time[1]. As a result, WRC-coated structures are periodically inspected for abrasion and breaches, which are then fixed by recoating the damaged areas. If the erosion is not detected in time, the resultant damage can be catastrophic. New methods are needed to accurately and precisely detect coating erosion in real time.

[1]Okanagan MicroElectronics and Gigahertz Applications Laboratory, School of Engineering, Faculty of Applied Science, University of British Columbia, Kelowna, BC V1V 1V7, Canada. [2]Department of Mechanical and Industrial Engineering, University of Toronto, Toronto, ON M5S 3G8, Canada. ✉e-mail: mohammad.zarifi@ubc.ca

Among the various methods of inspection, Non-Destructive Inspection (NDI) procedures are preferred for their potentially non-invasive, cost efficient, remote, and non-contact evaluation capabilities[7]. Testing systems that leverage NDI sensing methodologies include ultrasonic, eddy current, acoustic, and magnetic flux technologies[8]. The working principle, advantages, and drawbacks of commonly-used sensing modalities are summarized in Table S1[8-17]. The primary limitations of the listed sensing methodologies include the inability to detect erosive wear in real-time, and the difficulty in the remote monitoring of the system response. Additionally, NDI systems that use ultrasonic and magnetic particle inspection sensors are suitable for wear detection of only homogeneous coatings, whereas most industrial coatings are heterogenous and include at least a primer layer or bond-coat, mid-layer, and top-coat. Though unavailable today, an NDI system that could identify which layer was eroded, its wear depth, and the rate of erosion in real time and remotely would help ensure the safety and reliability of the coated structure. There remains a need to develop a contactless sensor that can remotely detect the erosive wear of multi-layer protective coatings in real-time.

Systems that leverage microwave-based sensors have demonstrated wide range of capabilities in the inspection and characterization of dielectric materials and composite structures in many essential applications[18]. The wave parameters of microwave sensors, including operational frequency, amplitude, phase, and polarization, are easily optimized for specific use cases[18]. Among the various classes of microwave resonator sensors, split ring resonators (SRRs) are particularly interesting due to their planar structure, ease of fabrication, passive operation, and conformability on various surfaces. SRRs have previously been used for chemical[19-21] and biological applications[22-26], material characterization[27,28], and corrosion monitoring[29-31]. They have also been used for real-time sensing applications including ultra-violet light detection[32], ice sensing[33,34], and gas sensing[35]. Systems utilizing microwave sensors including SRRs and complementary SRRs have also been developed specifically for the detection of surface and sub-surface level crack and defect formation in coatings[36-39]. Additionally, in a series of works, Deif et al. used linear arrays of spiral radio frequency identification (RFID) sensors to detect and characterize the cause of corrosion and defects in pipeline coatings[40-42]. By monitoring the changes in the resonant parameters (mainly the resonant frequency), the sensor could quantify and distinguish between the various causes of coating damage (water ingress and air ingress)[43-46].

However, the limited adoption of microwave-based NDI systems for coating wear detection is mainly due to the requirement of specialized equipment and expertise, resulting in high operational costs for reliable measurements. Additionally, erosive wear is an intricate and complex process, often occurring in harsh environments, making it challenging to develop an effective microwave NDI system that can accurately assess wear conditions. The utilization of these systems in eroding environments is impeded by their susceptibility to interference, signal degradation, and the complexity of calibration procedures. Further, the various WRC materials can exhibit different electromagnetic properties, necessitating the development of a system that is adaptable to many different WRCs.

In this work, we develop a differential SRR-based NDI system for the real-time detection and monitoring of erosive wear in multi-layer coated surfaces. The WRC-coated microwave NDI system functions by monitoring the variations in the resonant parameters of the embedded SRRs as a function of wear depth of WRCs. Contrary to prior works, the system can detect and locate the eroding layer in multi-layered coatings in addition to detecting the total wear depth. The system consists of a smart wireless readout circuitry for the remote and autonomous monitoring of WRC erosive wear, eliminating the use of Vector Network Analyzers (VNAs). The robustness of the developed system is also verified by observing the system's response in different harsh environments, including various temperature, humidity, and ultraviolet light (UV) exposure conditions.

## Results and discussion
### Operational principle
The developed microwave-based NDI system operates by real-time monitoring of the resonant characteristics of the SRRs, enabling remote erosive wear detection of WRCs in applications such as wind turbine and helicopter blades (Fig. 1a). The resonant characteristics of the SRRs are primarily dependent on their length ($l$) and the effective permittivity of its surrounding homogeneous dielectric medium, $\varepsilon_{\text{eff}}$ (Fig. 1b), as per Eq. 1.[47]

$$f_{\text{r}} = \frac{v_0}{2l\sqrt{\varepsilon_{\text{eff}}}} \tag{1}$$

where $f_{\text{r}}$ is the resonant frequency of the SRR and $v_0$ is the speed of electromagnetic wave propagation in free space.

The SRR was mounted over a dielectric substrate (with relative permittivity $\varepsilon_{\text{sub}}$) and was electromagnetically excited using the microstrip transmission line (MTL) connected to a VNA (Fig. 1c). In a MTL configuration, a significant portion of the electromagnetic field lines were confined in the substrate while the remaining fields extended into the air media. This distribution resulted in an effective permittivity $\varepsilon_{\text{eff}}$ approximated by Eq. 2 for MTLs operating in the Quasi-TEM mode, an EM wave propagation mode in which electromagnetic field lines are not completely contained in the substrate:

$$\varepsilon_{\text{eff}} = \frac{\varepsilon_{\text{sub}} + \varepsilon_{\text{top}}}{2} + \frac{\varepsilon_{\text{sub}} - \varepsilon_{\text{top}}}{2\sqrt{1 + \frac{12h}{w}}} \tag{2}$$

Here $h$ and $w$ are the height of the substrate and width of the microstrip line, respectively, and $\varepsilon_{\text{top}}$ is the permittivity of the medium above the MTL. In the absence of media over the SRR (i.e., air), $\varepsilon_{\text{top}}$ is equal to the relative permittivity of air, and thus $\varepsilon_{\text{top}} = 1$. For a coated SRR, the effective permittivity ($\varepsilon_{\text{eff}}$) will depend on the relative dielectric permittivity of the coating ($\varepsilon_{\text{coat}}$) and its thickness ($t$). For sufficiently thick coatings, the electromagnetic fields over the SRR do not extend beyond the coating, and hence $\varepsilon_{\text{top}} = \varepsilon_{\text{coat}}$. However, for thinner coatings, the electromagnetic fields partly extend beyond the coating thickness and into the air. The top layer can therefore be considered as two sublayers consisting of the coating material and air media, with an $\varepsilon_{\text{top}}$ value approximated using a modified version of the method derived by Chen et al.[48] given by Eqs. 3 and 4.

$$\varepsilon_{\text{top}} = \frac{1 - 2G}{1 + G} \tag{3}$$

$$G = \frac{1 - \varepsilon_{\text{coat}}}{2 + \varepsilon_{\text{coat}}} \left(\frac{t}{z}\right)^3 \tag{4}$$

Here $z$ is the maximum height covered by the electromagnetic field in the direction normal to the SRR plane. In the case of a multi-layer WRC consisting of $n$ layers (Fig. 1b), the $\varepsilon_{\text{top}(n)}$ value can be obtained recursively using a modified version of Eqs. 3 and 4 in which $\varepsilon_{\text{top}(n-1)}$ is initially found at each step, using Eqs. 5 and 6[48],

$$\varepsilon_{\text{top}(n)} = \frac{1 - 2G_n}{1 + G_n} \varepsilon_n \tag{5}$$

$$G_n = \frac{\varepsilon_n - \varepsilon_{\text{top}(n-1)}}{2\varepsilon_n + \varepsilon_{\text{top}(n-1)}} \left(\frac{t_{n-1}}{t_n}\right)^3 \tag{6}$$

Here $\varepsilon_n$ is the relative permittivity of the individual layers and $t_n$ is the thickness from the SRR to layer $n$ (Fig. 1b). For both single- and multi-layer coatings, erosive wear reduces $\varepsilon_{top}$ since the electromagnetic field lines begin to interact with the air media. This results in a decrease in $\varepsilon_{eff}$, increasing the resonant frequency monitored by the developed system (Fig. 1d).

### Erosive wear detection of WRC

The wear detection capability of the developed SRR-based NDI system was first evaluated by the erosive wear of a 2.5 mm-thick epoxy coating. The WRC-coated system operated with a resonant frequency of 2 GHz, resonant amplitude of −53 dB and a −3 dB $Q$-factor of 37. Initially, to find the sensing threshold of the developed system the erosive wear of the epoxy coating was performed gradually with wear depths of ~50 μm. Figure 2a illustrates the erosive wear of the epoxy-coated SRR and its wear depth measurement (height maps) obtained using a 3D scanning microscope. The 50 μm wear depth facilitated the determination of the NDI system's operational range, observed to be ~2.2 mm in coating thickness for the epoxy-based WRC investigated. Furthermore, at a thickness of 2.2 mm, 50 μm of erosive wear of the WRC resulted in a resonant frequency increase of 1.4 MHz, due to the decrease in the effective permittivity in the medium, as per Eqs. 1 and 2. For the selected coating, at greater thicknesses, the material-

electromagnetic interaction was not significant enough to produce an observable change in the system's response.

Subsequent erosive wear cycles (with arbitrary wear depths) increased the system's operational parameters, primarily the resonant frequency and $Q$-factor (Fig. 2b). The increase in the $Q$-factor at greater wear depths can be attributed to reduced amount of losses that occur within the coating, with further erosion resulting in the reduction of energy dissipation within the SRR medium. Thus, the instantaneous values of the resonant frequency and $Q$-factor of the SRR (post calibration of the system to the specific WRC), enabled the real-time estimation of the wear depth. The sensitivity and resolution of the developed NDI system were determined from Fig. 2b by analyzing the rate of increase of the operational parameters as the wear depth increased. At a wear depth of 1500 μm, the SRR demonstrated a sensitivity of 4 MHz for every 30 μm of erosive wear (averaged). However, at a wear depth of 2000 μm, the sensitivity increased to about 22 MHz for every 30 μm of erosive wear. The increase in the SRR's sensitivity at greater wear depths was caused by the higher electromagnetic field intensities that interacted with the coating layer closer to the gap region of the SRR (Fig. S2). In addition to the instantaneous values, the dynamic variations of the system's operational parameters also facilitated the estimation and verification of the wear depth. It can be observed from Fig. 2b that the erosive wear of the WRC-coated

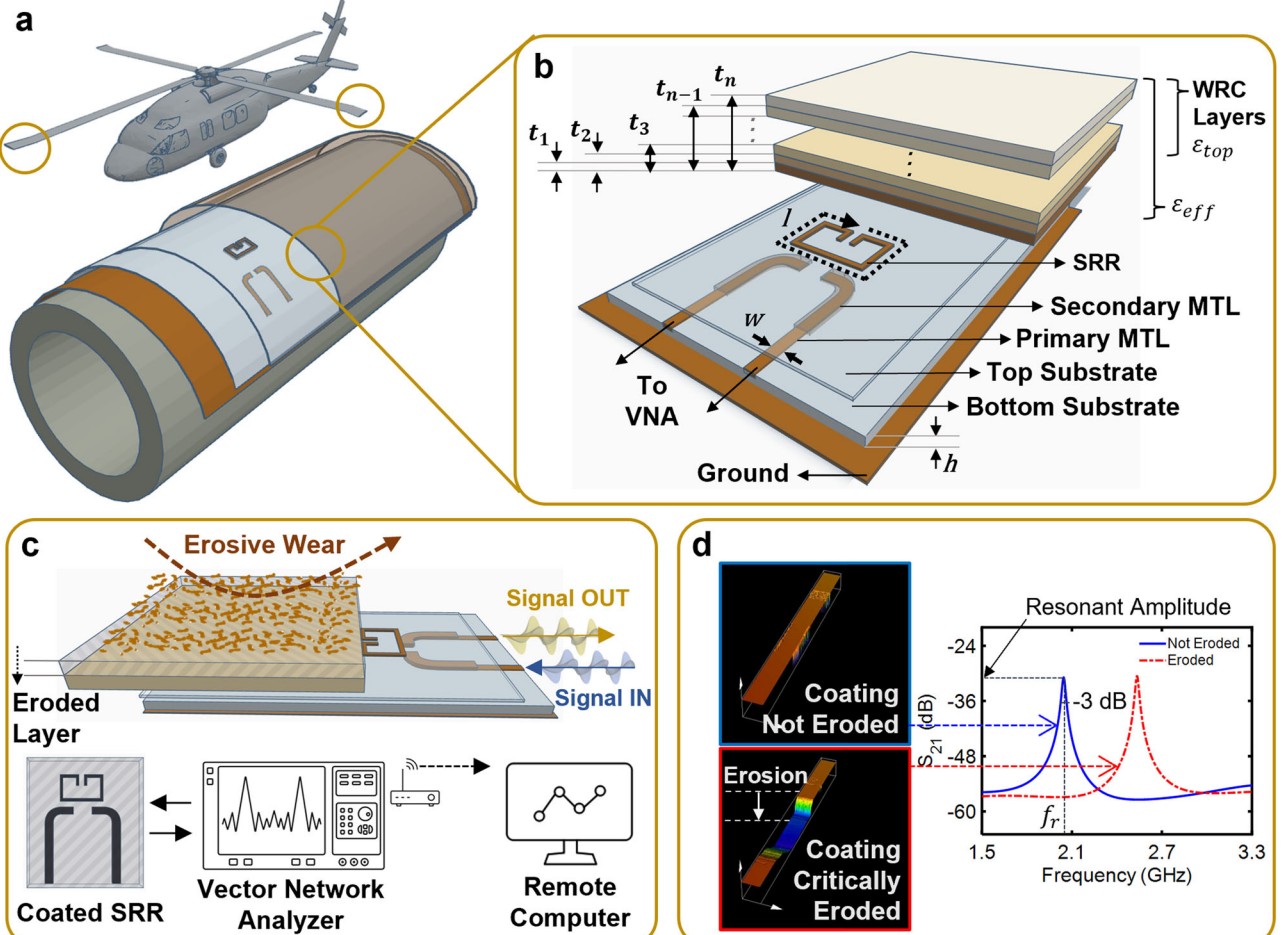

**Fig. 1 | Design and operational principle of the microwave SRR-based erosive wear detection system. a** An illustrated application of the developed system on the leading edges of helicopter blades and pipelines, **b** incorporation of the SRR-based system underneath the multi-layer WRC with its three-dimensional cross-sectional view depicting the components of the developed system and the thickness of WRCs above the SRR, **c** erosive wear of the WRC and the system's response observed using a Vector Network Analyzer and wirelessly communicated to a remote computer, and **d** height maps depicting the erosive wear of the WRC along with the corresponding system response. Observe the increase in the simulated resonant frequency as the WRC eroded, caused by the decrease in the effective permittivity. WRC wear resistant coating, SRR split ring resonator, MTL microstrip transmission line.

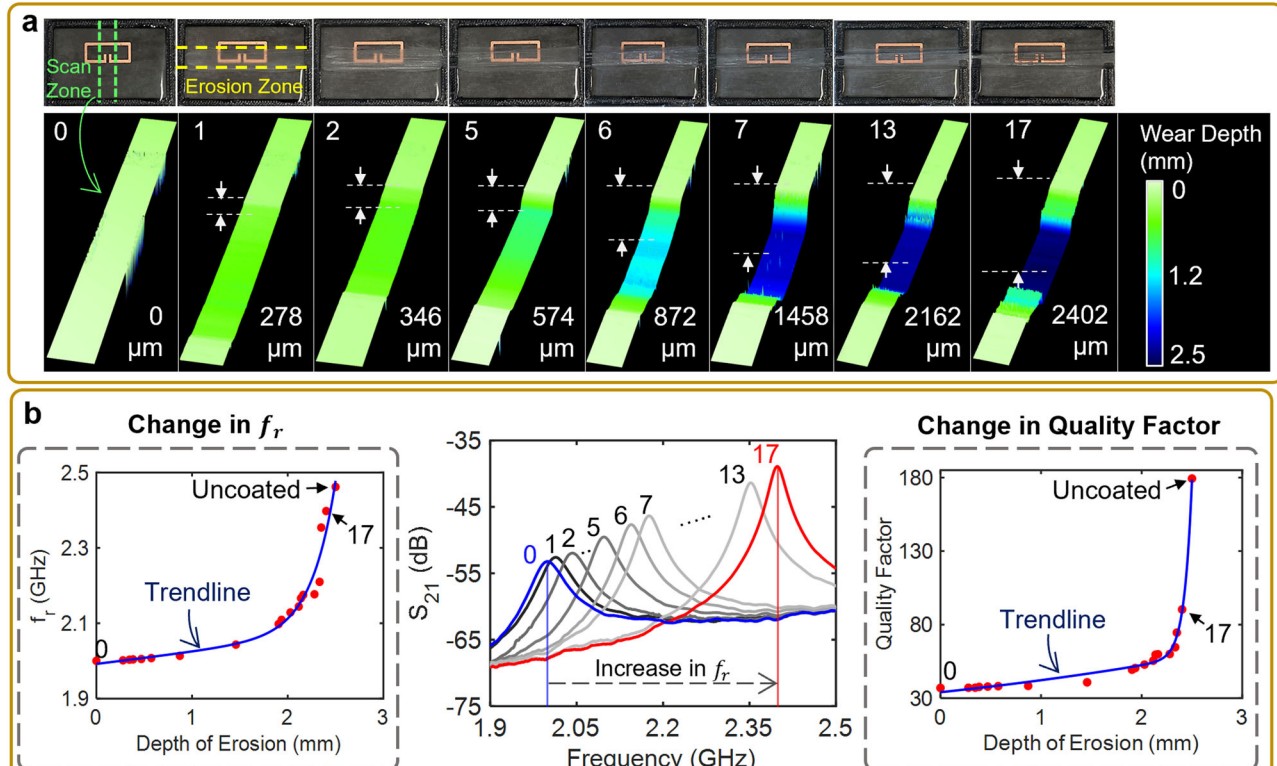

**Fig. 2 | Testing and performance analysis of the SRR-based erosive wear detection system. a** Erosive wear of the WRC-coated SRR-based NDI system and their corresponding height maps obtained using a 3D scanning microscope. The erosive wear of the WRC was performed around the erosion zone, over 17 cycles of manual wear. Seven representative height maps (taken along the scan zone) for erosion cycles 1, 2, 5, 6, 7, 13, 17, and the uneroded condition (Cycle 0) are shown. **b** Response of the developed system: resonant frequency, $S_{21}$, and $Q$-factor, to the erosive wear of the WRC over 17 wear cycles, along with the fitted trendlines. Each $S_{21}$ curve corresponds to the appropriate erosion cycle and wear depth as observed in (**a**). $f_r$ resonant frequency.

SRR-based NDI system lead to an increase in both the resonant frequency and the Q-factor values at an increasing rate.

From the operational perspective, the real-time detection and monitoring of erosive wear using the developed SRR-based NDI system involves a two-step process. Firstly, the system's calibration phase is initiated by mapping the system's wear response, specifically the resonant frequency and $Q$-factor, to the corresponding wear depth. Using the collected data, a mathematical relationship, often represented as a trendline or regression line, is established between the system's response (resonant frequency and $Q$-factor) and the wear depth. In the obtained results, the relationship between the wear depth and the system's response is modeled using a Gaussian function that can capture the decay of the electromagnetic field (from the SRR) in free space[49–52], given by:

$$d = a_1 * e^{-\left(\frac{x-b_1}{c_1}\right)^2} + a_2 * e^{-\left(\frac{x-b_2}{c_2}\right)^2} \quad (8)$$

where $d$ is the depth of erosion, $x$ is the system operational parameter and $a$, $b$, and $c$ refer to the amplitude, median, and standard deviation of the function, respectively. For the resonant frequency increase, the terms in Eq. 8 were $a_1 = 2.861e + 13$, $a_2 = 2.171$, $b_1 = 19.04$, $b_2 = 10.1$, $c_1 = 2.929$, $c_2 = 34.47$. For the $Q$-factor increase, the terms were $a_1 = 2.335e + 21$, $a_2 = 1.802e + 07$, $b_1 = 9.141$, $b_2 = 120.6$, $c_1 = 0.9966$, $c_2 = 33.23$ (Fig. 2b). The fitted trendlines resulted in $R$-squared values of 0.96 and 0.99, for resonant frequency and $Q$-factor, respectively. Following the calibration phase, the system is deployed for usage, during which it continuously captures the response of the coating and applies it to the pre-established trend lines. By comparing the real-time response of the system to the trend lines, the system can determine and estimate the wear depth of the coating as erosion occurs. The utilization of the trendlines serve to not only predict the coating thickness/wear depth, but also enhances the system reliability through data analysis and outlier detection. However, note that both the trendlines and the system performance metrics (sensitivity, resolution, and operational range) will depend on the specific WRC being monitored. Hence, proper calibration of the SRR-based system to the specific WRC prior to its installation and usage is necessary for the predictive analysis of the system's response.

**Compensating for external noise using a differential SRR system**
Erosive environments can also expose the WRC and SRR-based NDI system to extreme temperature and humidity variations as well as UV radiation. Harsh environmental conditions affect the performance of SRR-based microwave NDI systems[53,54], which would result in inaccurate wear depth measurements and reduced system resolution. To minimize environmental impacts, a differential mode SRR-based NDI system was designed and investigated by incorporating an electromagnetically shielded reference SRR that compensated for the system's response variations (Fig. 3a; see "Methods: Differential SRR-based system design").

The interference between the response of the reference SRR and the sensing SRR was reduced by designing the individual elements to operate at different frequencies (~1.6 GHz difference). The ability of the proposed design to shield the reference SRR was verified using cross-sectional electromagnetic field distribution plots, simulated using Ansys HFSS. The right hand side of Fig. 3a denotes the near-zero $E$-field intensity in the area over the copper shield, indicating that the reference SRR was electromagnetically shielded. The developed differential SRR-based NDI system was tested in four different environmental

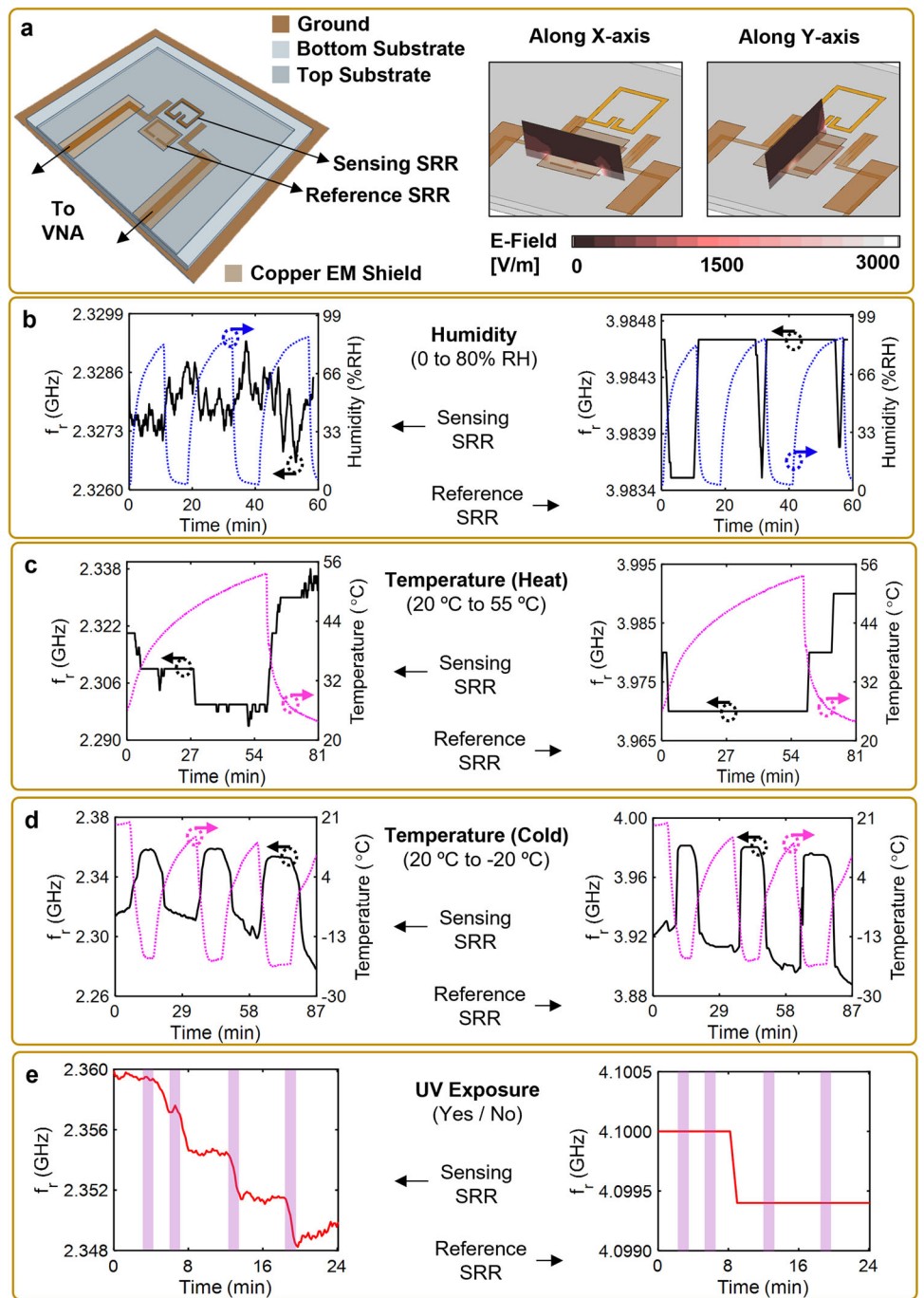

**Fig. 3 | Design of differential mode SRR-based NDI system and its performance analysis in various environments. a** Model of the proposed differential NDI system which incorporates a planar electromagnetically-shielded reference SRR, and electromagnetic field distribution simulations across the split ring gap of the reference SRR along the X-axis and Y-axis, verifying the shielding of the reference SRR. **b–e** Response of the developed differential system to varying **b** humidity, **c** temperature (heat), **d** temperature (cold), and **e** ultraviolet radiation exposure. In (**b**, **c**), the WRC is polyurethane, whereas in (**d**, **e**) it is polyethylene.

control chambers using two types of WRCs (see "Methods: Environmental impact test," and Figs. S4–S7) and the system's response was observed and is summarized in Table 1. The effect of humidity on the system's resonant frequency was minimal; when coated with a polyurethane-based WRC (typical used for aerospace and automotive applications), the sensing SRR varied by about 0.2 MHz while the reference SRR stayed constant (Fig. 3b). Similarly, a differential change of 1.5 MHz between the sensing SRR and the reference SRR was observed when the temperature was increased from 20 to 55 °C (Fig. 3c). However, in the absence of the differential topology, this change was recorded to be 15 MHz. The significantly reduced

1.5 MHz change was mainly characterized by the variations in the mechanical (thermal expansion) and the electrical (resistance) properties experienced by the system and the WRC, due to the temperature increase. For colder temperatures and UV exposure, a polyethylene-based WRC was used as a representative wind turbine blade coating. When evaluated from 20 to −20 °C, the differential value between the changes of resonant frequency for sensing and reference SRRs was as high as 4.5 MHz, due to the formation of a thin layer of frost in the air gap between the top and the bottom substrate (Fig. 3d). A difference of 10.5 MHz was observed when this system was exposed to UV radiation, which altered the dielectric properties of this specific WRC layer

**Table 1 | Variations in resonant frequency of the sensing and reference SRR with varying humidity, temperature, and UV exposure**

| Parameter | $f_r$ Variation (MHz) | | $\Delta f_r$ before exposure (GHz) | $\Delta f_r$ after exposure (GHz) | Change in $\Delta f_r$ (MHz) |
|---|---|---|---|---|---|
| | Sensing SRR | Reference SRR | | | |
| Humidity (0 to 80% RH) | 0.2 | 0 | 1.6568 | 1.6566 | 0.2 |
| Hot temperature (20 to 55 °C) | 15 | 13.5 | 1.6565 | 1.6550 | 1.5 |
| Cold Temperature (20 °C to −20 °C) | 39 | 34.5 | 1.6050 | 1.6095 | 4.5 |
| UV exposure (exposed/not) | 11 | 0.5 | 1.7405 | 1.7480 | 10.5 |

For the humidity and hot temperature tests, the WRC was a 3M polyurethane tape. For the cold temperature and UV tests, the WRC was ultra-high molecular weight polyethylene.

(Fig. 3e)[55]. Overall, it would be imperative to observe and calibrate the system's performance in varying environmental conditions prior to field installation and usage. Note, though, that the exact values recorded (Table 1) were specific to the selected WRCs and a different response would be observed if the differential SRR system were coated with an alternate WRC. However, the primary concern would be the variations in the system output readings and less on the absolute value of the readings. Thus, by leveraging an in-built differential unit, the system effectively prevented the effect of environmental loading of the sensing SRR and eliminated the need for an external reference tool.

### AI-enabled multilayer WRC erosion detection and estimation

Most real-world protective coatings consist of multiple layers that vary in composition, purpose, thickness, and application method. The individual layers of a multi-layer WRC vary from each other in material and dielectric characteristics and resemble a layered stack from a cross-sectional view. During the erosion of such multi-layer WRCs, it is thus critical to not only estimate the total wear depth but also determine which layer is being eroded and its erosion rate. To achieve this, the planar copper shield in the system design was extended from Port 1 to Port 2, providing further enhancement in terms of shielding both the reference SRR and the MTL (Fig. 4a), and preventing accidental loading of these elements. This design modification also mitigated the potential risk of the copper shielding (over the reference SRR) acting as a patch antenna. The developed system was coated with a representative multi-layer of three WRCs, comprising of polyethylene, polyimide, and polyurethane coatings each with a thickness of 0.6 mm (see "Methods: Experimental setup for erosion wear testing"; Fig. 4a). The sensing SRR of the multi-WRC coated system operated at a resonant frequency of 1.989 GHz with a resonant amplitude of ~−18 dB, while the reference SRR operated at 2.712 GHz with a resonant amplitude of ~−29 dB. The multilayer coating was eroded by stripping each WRC layer, while the $S_{21}$ measurements were recorded in real-time using the VNA (Fig. 4b). The resonant frequency of the sensing SRR increased with increasing wear depth, indicating a decrease in the effective permittivity caused due to material loss. On the one hand, the rate of resonant frequency change increased as each WRC layer was eroded, demonstrating the non-uniformity of the relative permittivity value between each WRC layer. On the other hand, the response of the reference SRR exhibited a negligible change in its resonant frequency, indicating that there were minimal fluctuations in the environmental parameters.

To perform a quantitative analysis of the observed system's response, the change in resonant frequency was plotted as a function of the wear depth (Fig. 4c). In the analysis, linear regression was executed for the erosive wear of each WRC layer with a thickness of 0.6 mm. The selection of linear fitting over the previously used Gaussian fitting was primarily due to the simplicity of interpretation and

ease of comparison. The erosive wear of the 0.6 mm thick top coating (polyurethane) resulted in a resonant frequency increase of 27 MHz, yielding a slope of 0.04 GHz/mm. Similarly, the wear of the middle coating (polyimide) increased the resonant frequency by a further 81 MHz, resulting in a slope of 0.14 GHz/mm. Finally, the wear of the bottom polyethylene coating caused a resonant frequency increase of 294 MHz at a rate of 0.49 GHz/mm.

To compare the system's response for erosive wear of homogeneous and heterogeneous WRCs, the experiments were repeated by subjecting a homogeneous layer of 1.8 mm-thick polyethylene to erosive wear. Similarly, the resonant frequency change was recorded, plotted against wear depth, and regression lines were calculated for every 0.6 mm of erosion. Figure 4d demonstrates the variable slopes of the system's response towards the erosive wear of homogeneous and heterogeneous WRCs. The resonant frequency of the 1.8 mm thick homogeneous WRC-coated system was greater than that of the 1.8 mm-thick heterogeneous WRC-coated sensor by ~38 MHz. The difference in the resonant frequency indicated a higher effective dielectric permittivity in the case of heterogeneous coating. As the top layer eroded (wear depth = 0.6 mm), the resonant frequency of the heterogenous coated system increased marginally higher than that of the homogeneously coated system, observed through the trendline slopes. The slope difference increased to 0.05 and was significant during the erosive wear of the middle layer (wear depth = 1.2 mm), indicating that the dielectric permittivity of the middle layer in the heterogenous WRC was greater than that of polyethylene. As the bottom layer was eroded (wear depth = 1.8 mm), there was no noticeable difference in the slopes because in both the scenarios the final 0.6 mm of the WRC coating was the same material (polyethylene). The results of this experiment indicate that by extracting the slope of the resonant frequency shift during the erosive wear of the multi-layer heterogeneous WRCs, the system can precisely identify the layer under erosive wear. Prior to installation and field usage, it is imperative to calibrate the system according to the specific WRC being utilized, whether it is homogeneous or heterogeneous. Calibration is necessary due to the material and dielectric property dependence of the system's response. Additionally, it was observed that the system's response differed when the order of the different WRC layers was interchanged. This was attributed to the fact that the E-field in the SRR exhibited variable interaction with different WRC layers owing to their distinct dielectric properties. This was expected, as altering the order of the WRC layers lead to variations in the total permittivity ($\varepsilon_{top}$) as depicted in Eqs. 5 and 6, consequently influencing the resonant frequency and the system's sensitivity.

In order to enhance the system's capability for precise coating monitoring and proactive maintenance planning, a predictive analytics model similar to a deep learning model was incorporated. To achieve this, a long short-term memory (LSTM)-based recurrent

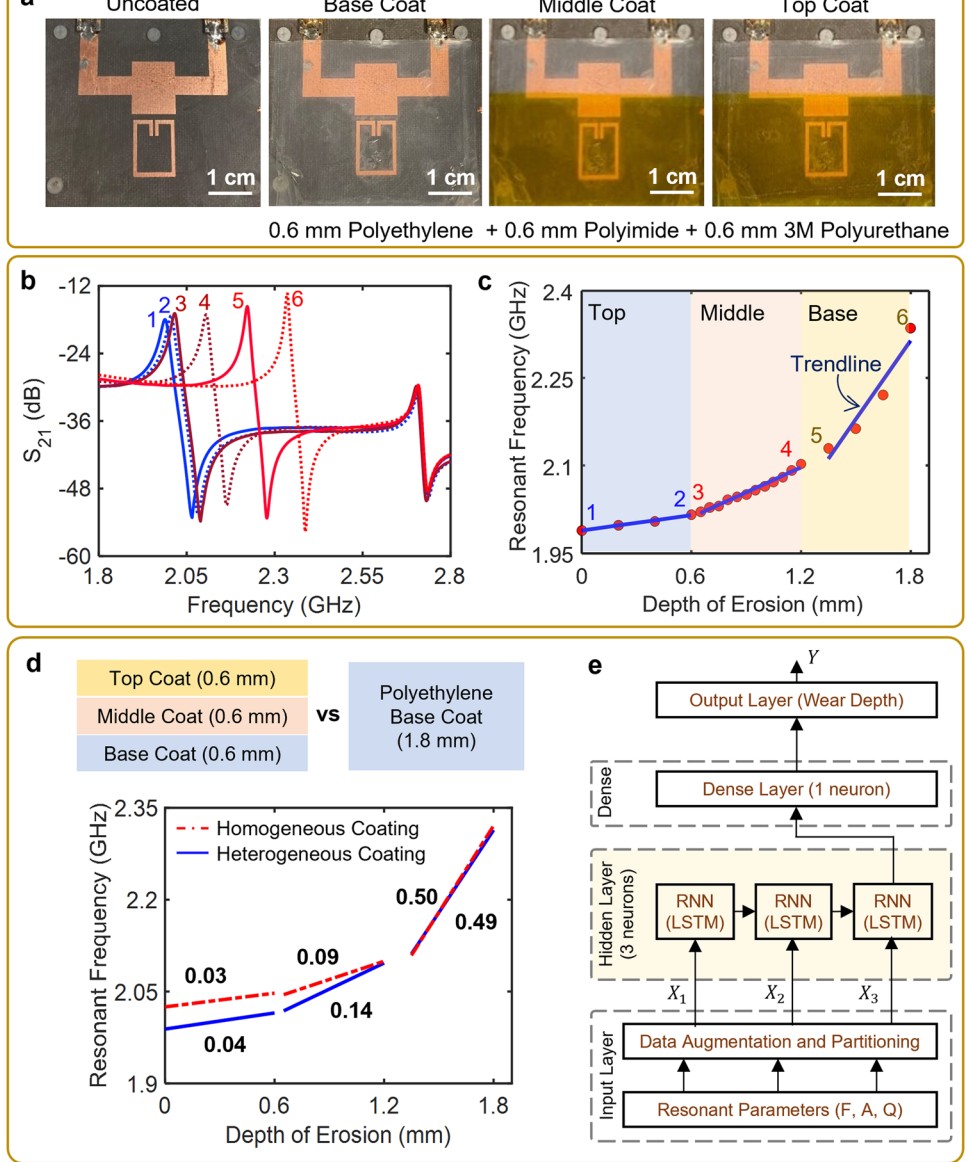

**Fig. 4 | Detection and positioning of erosive wear of multi-layer heterogeneous WRC coatings using the developed differential SRR system. a** Differential SRR NDI system coated with a multi-layer WRC coating, **b** response of the system to the erosive wear of the different WRC layers, **c** increase in the resonant frequency with increasing wear depth. Note the numbers in both (**b**, **c**) denote the beginning and the end of erosive wear of each layer, **d** comparison of the system's response to wear of homogeneous and heterogeneous coatings along with the slope value, and **e** predictive analysis of the wear depth using LSTM-based RNN model. RNN recurrent neural network, LSTM long short-term memory.

neural network (RNN) model was implemented since erosive wear is a dynamic process that evolves over time (see "Methods: Recurrent neural network modeling). For this application, RNNs are designed to model sequential data by maintaining an internal memory of past information (Fig. 4e). Using the system's response obtained from the erosive wear of the heterogeneous WRC, the dataset was augmented and split into training and test sets in the ratio 80:20. The test results demonstrated a mean squared error (MSE), mean absolute error (MAE), and root mean squared error (RMSE) of 0.0001, 0.008, and 0.01 respectively, validating that the predicted values were close to the actual values. The developed LSTM-based RNN model was accordingly capable of accurately predicting the wear depth based on the provided resonant parameter data, with a high R-squared value of 0.99. Overall, the incorporation of a LSTM-based RNN model enhanced the predictive analytics capabilities of the developed wear-detection system.

## System integration with smart monitoring readout circuitry

The implementation of the sensor in real-world applications would also benefit from the elimination of the VNA, enabling portability and cost-effectiveness. Consequently, a unique readout circuitry capable of monitoring the resonant amplitude at predefined frequencies was designed, effectively emulating the functionality of a VNA (Fig. 5a). The integration of the WRC-coated (polyethylene) differential SRR-based system to the smart readout circuitry enabled the monitoring of the system's response (see Methods: Design of Smart Monitoring System). Initially, the circuitry performed a frequency sweep ranging from 1.9 to 2.7 GHz. Subsequently, using the response obtained from the spectrum mode sweep, three frequency points were chosen in the vicinity of the resonant peak of the sensing SRR. The amplitude values at these frequency points were monitored utilizing the readout circuitry's discrete measurement mode (Fig. 5b). The variations in the amplitude values at each resonant frequency was observed and mapped using the

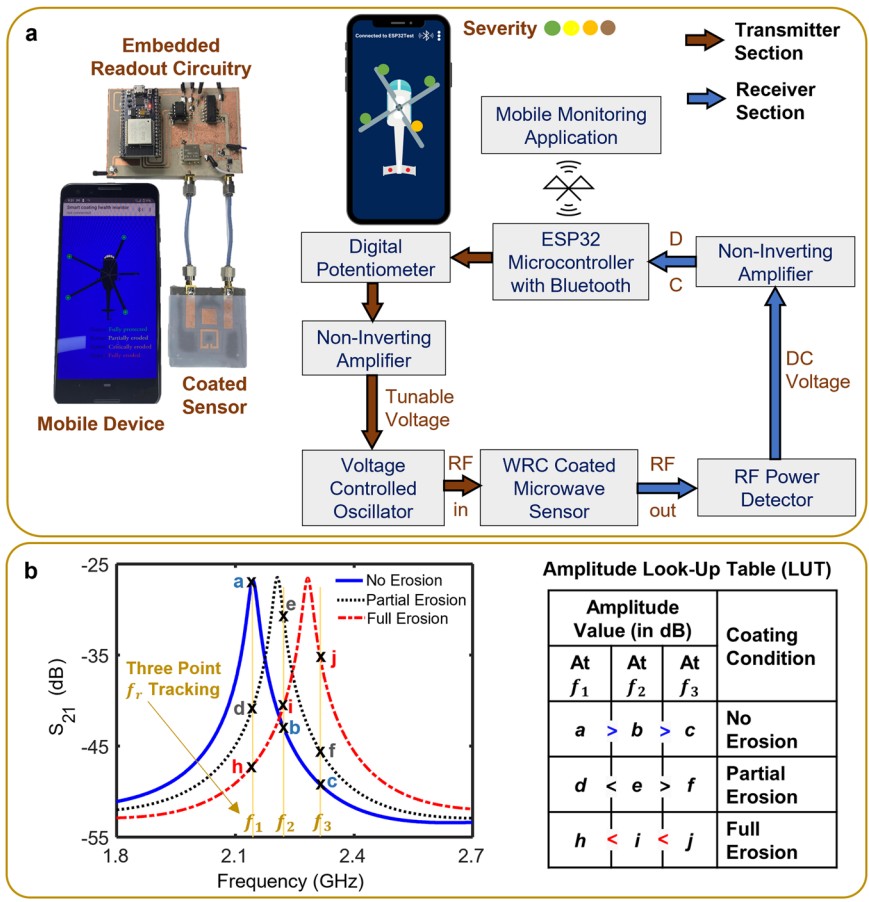

**Fig. 5 | Development and integration of a smart wireless readout circuitry to the differential SRR-based NDI system. a** Bluetooth-enabled smart readout circuitry connected to the WRC-coated differential NDI system, and the block diagram of the readout circuitry connected to the Android application via Bluetooth. **b** Amplitude monitoring mechanism at three discrete frequency points along with the amplitude look-up table to determine the erosion status of the WRC-coated differential SRR system. RF radio frequency.

look-up table as seen in the right-hand side of Fig. 5b. The data collected from the system was processed and transmitted wirelessly to a smart device using the built-in Bluetooth feature of the ESP32 microcontroller. As illustrated in Fig. 5a, the development of an in-house mobile application would allow users to monitor the erosion of the WRC and receive alerts accordingly.

While the presented system offers promising capabilities for WRC erosive wear detection, it is crucial to acknowledge its limitations and discuss potential avenues for improvement in future research. Firstly, the system's limited sensing range of 30 μm to 2.2 mm incapacitates its application to both thin film coatings as well as thicker coatings that are commonly used in the construction industry. A possible solution to this issue involves the optimization of the SRR's design parameters aimed to enhance its measurement range. Secondly, the inability of microwaves to penetrate metals renders the developed system unsuitable for wear detection in applications involving metal-based or conductive coatings. To address this limitation, the integration of the developed system with complementary NDI systems that leverage eddy current or magnetic flux sensors is required. Additionally, since the system's performance depends on the material properties of the coating, different coatings with varying compositions and characteristics could influence the system's response, potentially leading to reduced accuracy or reliability. Hence it is crucial to investigate techniques to mitigate the impact of material dependency and develop algorithms or calibration procedures to improve accuracy across different coating materials. Furthermore, the system's sensitivity to external interference from electromagnetic sources could affect the

reliability of measurements. Also, while the system offers real-time monitoring, potential delays, or limitations in wireless communication to remote devices could be encountered. Investigating shielding techniques or advanced signal processing algorithms to mitigate the impact of external interference, and optimizing the communication methods would ensure reliable and seamless monitoring in real-time. Cost considerations are also worth addressing since the implementation of the developed system requires embedding the planar SRRs beneath the coating layer. Exploring cost-effective alternatives for system integration methodologies could make the system more accessible and practical for widespread adoption. Thus, by critically assessing these limitations and addressing them in future research endeavors, the system's performance, applicability, and practicality could be improved, ultimately enhancing its utility for real-world coating erosion applications.

In summary, we investigated a differential-mode SRR-based microwave NDI system for the contactless and real-time detection of erosive wear in single and multi-layered coatings. The developed system operated at a resonant frequency of 2.33 GHz (sensing SRR) and 2.71 GHz (reference SRR). Through erosive wear experiments conducted over a 2.5 mm thick epoxy coating, the system demonstrated an average (linear) sensitivity of 13 MHz change of the resonant frequency for every 30 μm of wear depth. In addition, for the selected WRC, the system demonstrated a measurement range of ~2.2 mm. This indicates that the system's sensitivity was 1/73 of its measurement range, giving room for further exploration to improve the measurement range. The incorporation of the reference SRR to the developed

NDI system effectively compensated for the system's response variations caused due to changing environmental factors. Additionally, the differential microwave NDI system was tested to detect and locate the eroding layer and erosion rate for multi-layer WRCs, in addition to the total wear depth. Analyzing the variations in the first derivative of the system's resonant frequency change provided insights into the specific eroding WRC layer. Furthermore, the integration of AI-enabled SRR-based system to the custom-designed smart monitoring system allowed for real-time and remote monitoring of coating wear, showcasing its practical applicability in real-world applications including helicopter and turbine blades, gas pipelines, and naval infrastructure.

## Methods

### Design and simulations of SRR sensors

For the effective detection and monitoring of WRC erosive wear, the microwave NDI system consisting of a SRR was designed using Ansys High-Frequency Structure Simulator (HFSS). The system was intended to operate at the Industrial, Scientific, and Medical (ISM) frequency band of 2.4–2.5 GHz. To realize the requirement, as per Eq. 1, the SRR was designed with a physical length of 54 mm, a width of 1 mm, and a split ring gap of 1 mm. Similarly, the MTL used for the electromagnetic excitation of the SRR was designed with a physical length of ~66 mm and width of 1.5 mm. The design parameters of the SRR-based system were selected to optimize its sensitivity to the erosive wear of WRCs (see Supplementary Information Figs. S1–S3). Similar to prior works, these parameters were optimized by performing parametric optimization sweeps using Ansys HFSS[25,39]. The SRR and the MTL were designed on a Rogers 5880 substrate with permittivity of 2.2, dielectric loss tangent of 0.0009, and thickness of 0.79 mm, allowing for a flexible structure that could conform to non-planar surfaces. Furthermore, to reduce the likelihood of system damage due to harsh environments and to reduce the operational costs, the MTL (bottom) and the SRR (top) were designed on two different substrates and assembled one over the other (Fig. 1b). Hence, in the event of system damage, only the top substrate containing the SRR would need to be replaced. Additionally, to ensure proper alignment of the top and the bottom substrate, a secondary MTL, with a width twice that of the primary MTL, was added to the bottom surface of the top substrate (Fig. 1b).

### Experimental setup for erosion wear testing

The response of the developed SRR-based system to detect the erosive wear of WRCs (including homogeneous and multi-layer heterogeneous WRCs) was obtained by utilizing a VNA through which the $S_{21}$ response was observed. The VNA was calibrated using a Keysight Cal Kit 85521A calibration kit which ensured the repeatability in the system's response. The intermediate frequency (IF) bandwidth of the VNA was set to 500 Hz to enhance the signal-to-noise ratio of the measurements, with 2001 number of sweeping points and a power level of −5 dB. In the case of homogeneous WRC erosion testing, the SRR on the top substrate was coated with 2.5 mm-thick industry-grade Bisphenol A and Epichlorohydrin epoxy resin (permittivity of 2.7 at 2.5 GHz, estimated using Ansys HFSS simulations). The resin was cured for 48 hours at standard room temperature and pressure. The process of accelerated erosive wear of the WRC coated system was performed by rectilinear erosive motion of a mechanical filer over the WRC layer. The variations in the resonant parameters of the system's response, as the WRC eroded, were observed in real-time using a S5065 two port VNA. While the erosion was monitored in real-time, after 17 different amounts of erosion, each denoted as an erosion cycle, the WRC-coated sensor was disconnected from the VNA to determine the depth of erosion and calibrate the system's response. Height maps were then obtained using Olympus LEXT OLS5000 3D scanning microscope. In the height map scans, specifically along the scan zone as depicted in Fig. 2a, the eroded regions exhibited a high degree of surface roughness due to the physical wear process. Thus, the heights were averaged in the eroded regions to determine the average wear depth. The erosive wear experiment was carried out for 17 cycles until the epoxy layer was completely eroded away. Similarly, in the case of erosive wear of heterogeneous WRCs, the system was coated with three layers of WRCs, representing a multi-layer coating. The selected WRCs included multiple sheets of polyethylene coat (3 sheets of 200 μm thickness each), polyimide coat (12 sheets of 50 μm thickness each) and polyurethane protection film (4 sheets of 150 μm thickness each). The wear testing process involved the meticulous stripping of each WRC sheet while recording the system's response in real-time using a VNA.

### Differential topology design

The design of a differential SRR-based NDI system required the incorporation of a reference element (operating at 3.9 GHz) with a response that was independent from that of the sensing element. However, during the design of the differential system, the resonant characteristics of the sensing and the reference SRRs exhibited unequal variations to the erosive wear of WRCs, caused due to the non-uniform loading by the coating layer. The primary objective of the reference SRR was to detect and compensate for the response variations due to the environmental factors, and not due to the effect of the material sample. The challenge was tackled by developing a electromagnetically shielded differential mode NDI system in which the reference SRR and the MTL were shielded by a 35 μm-thick copper layer (Fig. 3a). Microwaves cannot penetrate through the copper shield and instead induce surface electrical currents that result in full reflection on incidence, interfering with electromagnetic waves. At the operational frequency of the reference SRR, the skin depth of the copper shield was about 1 μm indicating that the shield plate was equivalent to about 35 skin depths thick. The magnitude of the electromagnetic wave would have decayed to $e^{-35}$ of its initial value, thus having negligible interaction with the material placed over the copper shield, thus preventing accidental loading of the reference SRR. Therefore, the response of the reference SRR remains unaffected by changes in coating thickness and response of sensing SRR, making it primarily dependent on variations in environmental parameters.

### Environmental impact test

The differential NDI system was tested at varying temperatures (hot and cold), humidity, and UV exposure, thus mimicking several harsh environments. The developed system was coated with two different WRCs and its response was monitored in four different types of experimental setups that performed the desired environmental parameter variations. The different WRCs were selected to be representative in the test environment evaluated (for example, de-icing coatings in sub-zero temperatures), and to demonstrate the system's versatility to detect erosive wear of different coatings.

**Humidity variations.** The system was coated with 230 μm-thick 3 M Polyurethane Protective Tape 8544 (a wear-resistant WRC widely employed on aircraft and wind turbine blades). The coated system was placed in an airtight chamber equipped with moisture and air inlets from a bubbler, a fan for circulation, and a real-time relative humidity and temperature data logger (Fig. S4). The relative humidity varied from 0% RH to 80% RH and was controlled using Alicat Mass Flow Controllers. The system's $S_{21}$ response was recorded every 10 s by taking the difference between the two SRR's resonant frequencies ($\Delta f_r$) using an N5222B VNA for over 1 h through three cycles.

**Hot temperature variations.** To evaluate the effect of temperature (heat), the 230 μm-thick 3 M Polyurethane coated system was placed in an airtight temperature-controlled chamber (Fig. S5), with temperature varying from room temperature to over 50 °C. The $S_{21}$

measurements ($\Delta f_r$) were taken using the N5222B VNA, at the same time intervals of 10 s, over 80 min. During this period, the temperature in the chamber with a relative humidity of 56% RH increased from 26–53.7 °C.

**Cold temperature variations.** The developed system was coated with a 125 μm-thick ultra-high molecular weight polyethylene film (a known de-icing coating). To observe its response to cold temperatures, the coated system was placed over a Peltier plate with temperature varying from room temperature to −20 °C (Fig. S6). The $S_{21}$ measurements ($\Delta f_r$) were taken for 90 min using a hand-held Keysight N9918A VNA.

**UV exposure.** The 125 μm-thick polyethylene-coated system was placed in a UV chamber where the UV lamp emitted radiation for 1 min, followed by a variable observation/cooling period (Fig. S7). The $S_{21}$ measurements ($\Delta f_r$) were taken using a Copper Mountain S5065 2-Port VNA, and the system's response was recorded using LabView at 10 s time intervals.

### Recurrent neural network modeling

The integration of the developed system with a RNN model to enable predictive analysis of wear depth required a robust AI model capable of dealing with small data sets. The initial dataset consisted of 20 obtained values for resonant frequency, resonant amplitude, and quality factor, each corresponding to a particular wear depth. The dataset was expanded to 400 by adding augmented data points in order to improve the model's robustness and later normalized on a scale of 0 to 1. The augmented dataset was split into two parts: the training set and the test set, in the ratio 80:20. The training set was used to train the RNN model while the test set was used to evaluate its performance. The RNN model utilized Keras with TensorFlow and featured a single LSTM layer (3 neurons) followed by a dense layer (1 neuron) and the output layer (Supplementary Code 1). LSTM was ideal for sequential data, and the linear activation function aided to predict wear depth accurately. The selection of 3 neurons for the LSTM layer helped to strike a balance between capturing the required complexity of the problem while considering computational efficiency. The model was compiled utilizing an Adam optimizer and was trained for 100 epochs with a batch size 3, thus minimizing the loss between predicted and actual wear depths. The performance of the trained model was evaluated on the test set using metrics, including MSE, MAE, RMSE, and $R$-squared ($R^2$) scores.

### Design of smart monitoring system

A readout circuitry was designed comprising of a ROS-3800-119R voltage controlled oscillator (VCO), MCP4141 digital potentiometer, non-inverting amplifier using LM324AN operational amplifier, and a AD8314ARMZ RF power detector (Fig. 5a). The VCO operated between the frequencies 1.9 and 3.7 GHz, while the RF power detector operated from 100 MHz to 2.7 GHz. The custom-made smart monitoring read-out circuitry was capable of operating in two measurement modes: discrete and spectrum. In discrete measurement mode the circuit tuned the VCO to specific frequencies and measured the system's response at those discrete frequencies. Whereas in spectrum measurement mode the circuit performed a frequency sweep across a range of frequencies and measured the system's response over the entire frequency range. The implementation of the dual-mode system facilitated the dynamic selection of specific frequencies, enabling the monitoring of the coated system's response (amplitude values) at the chosen frequency points (Fig. 5b).

### Reporting summary

Further information on research design is available in the Nature Portfolio Reporting Summary linked to this article.

## Data availability
Source data are provided with this paper.

## Code availability
The code employed in the subsection "Recurrent neural network modeling" is provided as Supplementary Code 1.

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

## Acknowledgements

The authors thank the Syilx Okanagan Nation for the use of their unceded territory, the land on which the research was conducted. They acknowledge the support from Rogers Corporation for substrate donations and CMC Microsystems for providing software licenses. The authors acknowledge the financial support from the Department of National Defense of Canada (Contract No: IDEaS-CFP5-1A-CH 3-CP-5325) and the support from the Natural Sciences and Engineering Research Council of Canada (NSERC), through grant RGPIN-2018-04288 and the Canadian Foundation for Innovation (CFI) through grant nos. 38148 and 37904.

## Author contributions

M.H.Z. and K.G. conceived and supervised the project. V.B., O.N., and M.C.J. are responsible for theoretical design, and simulation. V.B. and M.C.J. performed the experimental testing of the system. V.B. and O.N. prepared and edited the manuscript including the figures. All authors discussed the results and contributed to the manuscript.

## Competing interests

The authors declare no competing interests.
