## [Peer Review File · Nature Communications]

Non-destructive erosive wear monitoring of multi-layer coatings using AI-enabled differential split ring resonator based systemREVIEWER COMMENTS

Reviewer #1 (Remarks to the Author):

In this paper, a differential planar microwave split ring resonators is designed to contactless and real-time monitoring the erosion of multilayer wear-resistant coatings. It is interesting topic. However, some issues should be discussed further.

1. As you mentioned in Table 1 that, the eddy current can detect surface or subsurface damages due to the skin depth. However, the skin depth can be adjusted by changing excitation frequency. therefore, it is also can be used in multiple structures. Such as the work in <https://doi.org/10.1080/10589759.2015.1081903> and in <https://doi.org/10.1109/TIM.2018.2872386>. The weak conductivity and permeability might be the reason. Further, what is meaning of coating lift-off in Table 1.

2. Two challenges for monitoring the erosion of multilayer wear-resistant coatings is proposed in section I as (i) the inability to detect erosive wear while the coated structure is in use (real-time sensing) and (ii) the difficulty in remote monitoring of the sensor data. So what are the performances of the proposed SRR in dealing with these two challenges?

3. Please clarify the contributions of this article in introduction part.

4. It is suggest to add a figure in 2.1 to illustrate the length of resonating structure, h and w are the height of the substrate and width of the microstrip line, to help the reader to understand the equations

5. There are 12 parameters in the microwave SRR, how to obtained them? And what is the considerations in seeking these parameters? Are they the optimal? Similarly, there are 8 parameters in Fig.3a, how to get them? Please explain the Fig.7C.

6. In Fig,6(a), what are the different line standing for? Please denote in the figure. In this figure, the resonant frequency is increasing with the cycles, however, it decreased for the last cycle, explain it please

7. The differential mode sensing can nullify the effect due to the environmental conditions. However, how to get the reference signal because the erosion condition is unknown in engineering. From Fig.7, we can see the structure of the sensing SRR and reference SRR is different, what is consideration? And how the difference influence the differential signal?

8. Why the coating thickness and material for hot Temperature Variations and Cold Temperature Variations is different. Please explain it

9. As a whole, too much content is involved in. it is suggested to reorganize according to your contributions. And the language should be concise further.

Reviewer #2 (Remarks to the Author):

The suggestion that THz sensors are expensive may be overstating the case, certainly MMICs are available nowadays from commercial foundries operating to 0.16 THz. The authors may want to check out Luna's Teramatrix offering? The response seems very sensitive to the SRR height above the sample. Could you comment on how this affects accuracy? Could you use higher order modes too?

Is the VNA calibrated? If so state how and if not why not - how do you guarantee repeatability?

The Q measured will also depend upon the conductor backing of the dielectric sample.

While the reference SRR conductor shield is presumably many skin depths thick, preventing field leakage this shield too has the potential to act as a patch antenna - did the authors consider this? Induced currents in the shield will radiate around the shield edges.

Some very well presented experimental work on environmental effects though.

Line 305: electric field not Electric.

The novelty of this approach appears to be in part the application where the only parameter being changed is the dielectric thickness indicating wear. Environmental changes are neatly compensated for through the reference resonator. Further the VNA is replaced by an analogue setup monitoring the resonant frequency of the SRR making for a compact sensor for use on helicopter blades. The significance of this work is relatively minor as a novel NDT technique but may have its own niche application.

Reviewer #3 (Remarks to the Author):

In the last paragraph of the Introduction, the authors indicated a novel differential sensor is introduced. However, what is novel about this differential sensor is not clear compared with previously published microwave differential sensors. Similarly, non-differential sensors have been published previously. So what is significant for the proposed technique? It would be good to expand this paragraph and give further convincing details.

What is the limitation of WRC total thickness in eqn 5 and 6 (for example, when the effective permittivity does not change)?

The theory in Section 2.1 was clear and easy to follow in terms of detecting multiple layers and defining their thickness and relative permittivities. However, in section 2.2 it is unclear what the permittivity of coated material and the erosive material was. Was the 2.5 mm coated layer verified with measurement? What about the erosive one?

This section needs substantial changes to clearly explain the steps for verifying and detecting multiple layers and erosive materials.

I assume erosion depth in figure 6 c and d are produced using eqn 8 considering f_r and Q factor? Need to make this point clear.

Section 3: it was mentioned, "Above it was demonstrated that microwave SRR sensors enable the real-time detection", however it was not real-time but post-processing.

Section 3.1: I suggest including the equivalent circuit for the differential sensor and producing simulation results for the circuit with verified EM simulation. Interesting to know particularly the effect of shielding plates.

The relation between the design theory of multilayer detection and the results presented in Section 3.2 is unclear (how this concept has been used to detect environmental changes). Or is it simply a single-layer measurement with changes in the dimension and permittivity of the single layer?

It would be good to present a table comparing the results from this work with state-of-the-art.

Reviewer #1 (Remarks to the Author):

1. As you mentioned in Table 1 that, the eddy current can detect surface or subsurface damages due to the skin depth. However, the skin depth can be adjusted by changing excitation frequency. Therefore, it is also can be used in multiple structures. Such as the work in <https://doi.org/10.1080/10589759.2015.1081903> and in <https://doi.org/10.1109/TIM.2018.2872386>. The weak conductivity and permeability might be the reason. Further, what is meaning of coating lift-off in Table 1.

Authors' Response:

Thank you for your comment. The mentioned works utilize eddy currents-based sensor to estimate the thickness of multi-layer metal coatings have been cited in the updated manuscript. The description of coating lift-off has been updated to coating detachment, which means the detachment of the coating layer from the structure for any reason, such as adhesive failure, water ingress, corrosion, etc.

The following changes are made in the modified version of the manuscript:

The mentioned works are cited, and the listed limitations of eddy currents sensor has been updated to just “Cannot detect coating detachment; Used mainly in metals”. This comparison table has also been moved to the Supplementary Information to maintain the focus on the performed work.

2. Two challenges for monitoring the erosion of multilayer wear-resistant coatings is proposed in section I as (i) the inability to detect erosive wear while the coated structure is in use (real-time sensing) and (ii) the difficulty in remote monitoring of the sensor data. So, what are the performances of the proposed SRR in dealing with these two challenges?

Authors' Response:

Thank you for the comment. Indeed, real-time, and remote sensing of WRC erosive wear are the two major challenges that are overcome using the proposed system.

First, all the measurements recorded using the SRR-based NDI system demonstrate real-time variations in its response with changes in the wear depth of the WRC layer. This can be seen in Figure 2, where erosive wear of ~ 2.5 mm thick epoxy-based WRC is monitored in real-time using the resonant parameters including resonant frequency and Q-factor of the SRR. These variations in these parameters were observed in real-time while the SRR-based system was still connected to the VNA which recorded the system's response during the erosive wear process.

Second, the portability, cost-effectiveness and the remote wireless monitoring of the system response are enhanced through the system's integration to a smart monitoring circuitry connected via Bluetooth to a remote mobile device. Much of this is shown in relation to Figure 5.

The following updates are made in the modified version of the manuscript:

Results: Erosive Wear Detection of WRC

“Thus, the instantaneous values of the resonant frequency and Q-factor of the SRR (post calibration of the system to the specific WRC), enabled the real-time estimation of the wear depth.”

Results: System Integration with Smart Monitoring Readout Circuitry

“The implementation of the sensor in real-world applications would also require the elimination of the VNA, enabling it to be portable, and cost-effective. Consequently, a dedicated readout circuitry capable of monitoring the resonant amplitude at predefined frequencies was designed, effectively emulating the functionality of a VNA (Fig. 5a). The integration of the WRC-coated (polyethylene) differential SRR-based system to the smart readout circuitry enabled the monitoring of the system's response (see Methods: Design of Smart Monitoring System). Initially, the response of the coated system was observed by operating the readout circuitry in the spectrum mode, wherein the circuitry performed a frequency sweep ranging from 1.9 GHz to 2.7 GHz. Subsequently, using the response obtained from the spectrum mode sweep, three

frequency points were chosen in the vicinity of the resonant peak of the sensing SRR. The amplitude values at these frequency points were monitored utilizing the readout circuitry's discrete measurement mode (**Fig. 5b**). The variations in the amplitude values at each resonant frequency was observed and mapped using the look-up table as seen in the right-hand side of **Fig. 5b**. The data collected from the system was processed and transmitted wirelessly to a smart device using the built-in Bluetooth feature of the ESP32 microcontroller. As illustrated in **Fig. 5a**, the development of an in-house mobile application would allow users to monitor the erosion of the WRC and receive alerts accordingly.”

3. Please clarify the contributions of this article in the introduction part.

Authors' Response:

Thank you for the comment. The primary contributions of the developed wear detection system include:

- a. Contactless and Non-Destructive Inspection of WRC erosion using a SRR-based microwave NDI system that can estimate the wear depth based on the variations in the resonant parameters of the SRR in real-time.
- b. Compensation of the impact of environmental noises on the system's response through the incorporation of an electromagnetically shielded reference SRR element, thereby improving its robustness and usage in harsh environments.
- c. AI-enabled detection of erosive of multi-layered heterogeneous coatings using the developed differential SRR-based system, with the capability to locate the layer of erosion, thus demonstrating its potential for real-world applications involving multi-layered coatings.
- d. Implementation of a smart readout circuitry to promote the portability, cost-effectiveness, and the remote wireless communication capabilities of the developed wear detection system.

The following paragraph has been added to the Introduction section to better clarify the contributions of the work:

“However, the limited adoption of microwave-based NDI systems for coating wear detection is mainly due to the requirement of specialized equipment and expertise, resulting in high operational costs for reliable measurements. Additionally, erosive wear is an intricate and complex process, often occurring in harsh environments, making it challenging to develop an effective microwave NDI system that can accurately assess wear conditions. The utilization of these systems in eroding environments is impeded by their susceptibility to interference, signal degradation, and the complexity of calibration procedures. Further, the various WRC materials can exhibit different electromagnetic properties, necessitating the development of a system that is adaptable to many different WRCs. In this work, we develop a differential SRR-based NDI system for the real-time detection and monitoring of erosive wear of multi-layer coated surfaces. The WRC-coated microwave NDI system functions by monitoring the variations in the resonant parameters of the embedded SRRs as a function of wear depth of WRCs. Contrary to prior works, the system can detect and locate the eroding layer in multi-layered coatings in addition to detecting the total wear depth. The system consists of a smart wireless readout circuitry for the remote and autonomous monitoring of WRC erosive wear, circumventing the need for a Vector Network Analyzer (VNA). The robustness of the developed system is also verified by observing the system's response in different harsh environments, including various temperature, humidity, and ultraviolet light (UV) exposure conditions.”

4. It is suggest to add a figure in 2.1 to illustrate the length of resonating structure, h and w are the height of the substrate and width of the microstrip line, to help the reader to understand the equations

Authors' Response:

Thank you for your suggestion. In the modified version, we have added these dimensions.

The following changes are made in the modified version of the manuscript:

In the revised Fig. 1b, the length of the SRR, height of the substrate and the width of the microstrip line are visually represented to enable better understanding of the equations. Additionally, Fig. 1 is brought to the front to provide an overview of what the section discusses.

5. There are 12 parameters in the microwave SRR, how to obtain them? And what are the considerations in seeking these parameters? Are they the optimal? Similarly, there are 8 parameters in Fig.3a, how to get them? Please explain the Fig.7C.

Authors' Response:

Thank you for the comment. The 12 parameters that were listed in Fig. 2b (previous version) are the design parameters. These parameters are non-critical but were selected to improve the sensitivity of the SRR system towards the erosive wear of the WRC layer. It is conventional to optimize the design parameters, specific to the application, through parametric optimization sweeps performed using electromagnetic modelling and simulation software such as Ansys HFSS, and that was done in this work. However, the more critical parameters are the length of the SRR and the permittivity of the media, which dictate the operational frequency as per Eq. 1 and 2 as mentioned in the manuscript. In the updated version, the Fig. 2b has moved to Supplementary Information Fig. S3.

Fig. 3a (previous version) represents the equivalent electrical lumped circuit model of the developed SRR-based system. The circuit parameters are approximated and calculated using the physical dimensions of the SRR and the MTL. They are further verified using the comparable approximations made in other works as cited in Supplementary Information (citations 11 and 12).

Fig. 7c (previous version) represents the electromagnetic field distribution along the XZ-plane. The intention of performing the simulation is to confirm the minimized interaction between the penetrated electromagnetic fields and the coatings. Based on the results Fig. S2 (updated version), the reduced intensity of the electric fields over the copper shield indicates the effectiveness of the copper shielding. Fig. S2 has been updated with a detailed description in the figure caption.

The following changes are made in the modified version of the manuscript:

In the *Methods: Design of SRR-based NDI system* section, the following brief information has been added: “The design parameters of the SRR-based system were selected to optimize its sensitivity to the erosive wear of WRCs (see Supplementary Information Fig. S1-S3). Similar to prior works, these parameters were optimized by performing parametric optimization sweeps using Ansys HFSS^{23,32,38,55}. The SRR and the MTL were designed over a Rogers 5880 substrate with permittivity of 2.2, dielectric loss tangent of 0.0009, and thickness of 0.79 mm, allowing for a flexible structure that could conform to non-planar surfaces. Furthermore, to reduce the likelihood of system damage due to harsh environments and to reduce the operational costs, the MTL (bottom) and the SRR (top) were designed on two different substrates and assembled one over the other (**Fig. 1b**). Hence, in the event of system damage, only the top substrate containing the SRR would need to be replaced. Additionally, to ensure proper alignment of the top and the bottom substrate, a secondary MTL, with a width twice that of the primary MTL, was added to the bottom surface of the top substrate (**Fig. 1b**).”

In the *Supplementary Information* document, the following information has been added:

“The circuit model and the parameters were designed and validated through simulations using Advanced Design System (ADS) and further confirmed through comparison with similar approximations performed in prior works^{11,12}.”

6. In Fig,6(a), what are the different lines standing for? Please denote in the figure. In this figure, the resonant frequency is increasing with the cycles, however, it decreased for the last cycle, explain it please

Authors' Response:

Thank you for these questions. We apologize for the confusion caused regarding the significance of the different lines represented in Fig. 6a (previous version). In Fig 6a (previous version), each line represented the response of the SRR-based system as the wear depth increased with each cycle of erosion, given in the color bar under Fig. 6b.

As for the decrease observed for the last cycle, there was ~1 dB difference in the resonant amplitude value between the last two cycles. The primary reason for this difference is that the last cycle of erosion completely removed the coating, whereas for the prior cycle the WRC remained, with an erosion depth of about 2.4 mm (wear cycle 17). Accordingly, the response of the system with and without the coating will be quite different. However, note that the two primary parameters of interest are the resonant frequency and Q-factor and not the resonant amplitude. In the updated manuscript, Fig. 6a (previous version) has been updated to Fig. 2 (updated version, also attached here as Fig. R1) for better understanding.

The following changes are made in the modified version of the manuscript:

Fig. 6 (previous version) has been replaced by Fig. 2 (modified version). In this figure, for better understanding, the subsets of each heightmap are numbered by their erosion cycle (Fig. 2a), and the corresponding system's response curves are matched to the heightmaps (Fig. 2b). We believe that this would improve the clarity of the Figure.

Fig R1. Testing and Performance Analysis of the SRR-based erosive wear detection system. (a) Erosive wear of the WRC-coated SRR-based NDI system and their corresponding height maps obtained using a 3D scanning microscope. The erosive wear of the WRC was performed around the erosion zone, over 17 cycles of manual wear. Seven representative height maps for erosion cycles 1, 2, 5, 6, 7, 13, 17 (completely eroded), and the uneroded condition (Cycle 0) are shown. **(b)** Response of the developed system: resonant frequency, S_{21} , and Q-factor, to the erosive wear of the WRC over 17 wear cycles, along with the fitted trendlines. Each S_{21} curve corresponds to the appropriate erosion cycle and wear depth as observed in Fig. 2a.

7. The differential mode sensing can nullify the effect due to the environmental conditions. However, how to get the reference signal because the erosion condition is unknown in engineering. From Fig.7, we can see the structure of the sensing SRR and reference SRR is different, what is consideration? And how the difference influence the differential signal?

Authors' Response:

Thank you for your question. As mentioned, erosion can be complex and unpredictable. However, it is important to build a robust NDI system that can detect erosive wear of WRC at all operational environments and additionally incorporate a reference element that can compensate for the environmental effects on the system's response. By incorporating an independent reference SRR with an operational frequency (3.9 GHz) which is 1.6 GHz higher than that of the sensing SRR (2.3 GHz), we are effectively isolating them so that they do not overlap. Since the reference SRR is electromagnetically shielded by the copper shields, any variations in the coating thickness caused due to erosive wear would have no impact on the response of the reference SRR.

The geometry of the reference SRR was modified to maintain a Q value similar to that of the sensing SRR to maintain similar and comparable variations in both their responses due to the environmental impact.

The following statement are added in the modified version of the manuscript for better clarity:

Results: Compensating for Environmental Noises using a Differential SRR system

“The response of the reference SRR was isolated from the sensing SRR by designing a significant operational frequency difference (1.6 GHz) between them, ensuring no interference between their respective responses.”

Method: Differential SRR-based System Design

“The design of a differential SRR-based NDI system required the incorporation of a reference element (operating at 3.9 GHz) with a response that was independent from that of the sensing element.”

“Therefore, the response of the reference SRR remains unaffected by changes in coating thickness and response of sensing SRR, making it primarily dependent on variations in environmental parameters.”

8. Why the coating thickness and material for hot Temperature Variations and Cold Temperature Variations is different. Please explain it.

Authors' Response:

Thank you for the question. While we could have used a single WRC throughout our work, we wanted to demonstrate the versatility of the developed system. We selected two WRCs that either find use in hot or cold environments. The 230 μm thick 3M Polyurethane Protective Tape 8544 is representative of wind turbine leading edge protective coatings, which often operate at ambient or warm temperatures. Alternatively, the 125 μm thick Ultra High Molecular Weight Polyethylene (UHMWPE) WRC was selected because it has been shown to be a promising coating to protect against icing in cold environments (see Azimi Dijvejin et al., *Nature Communications* **13**, 5119 (2022)). The unique electromagnetic properties for every coating will result in a differing system response during erosive wear. Hence, it is beneficial to build an NDI system that can operate effectively with multiple types of WRCs, as demonstrated by this part of our work.

The following brief statement is added to the *Methods: Environmental Impact Test* section:

“The different WRCs were selected to be representative in the test environment evaluated (for example, de-icing coatings in sub-zero temperatures), and to demonstrate the system's versatility to detect erosive wear of different coatings.”

9. As a whole, too much content is involved in. it is suggested to reorganize according to your contributions. And the language should be concise further.

Authors' Response:

Thank you for the suggestion. The manuscript has been majorly revised and reorganized to maintain the flow of presentation and now focuses on the primary contributions of the work, while being as concise as possible. A Supplementary Information has been created to aid with this, which contains the following parts of the work:

- Table 1 (previous version), which compares the benefits and drawbacks of different NDI systems (now Table S1)
- Figure 3 (previous version), which includes the design optimization, electromagnetic field simulations, and electrical lumped circuit model analysis of the developed SRR-based microwave NDI system (now Figs S1 and S3)
- Figure 5 (previous version), which shows the electric field distribution simulations (now Fig S2)
- Parts of Figure 8 (previous version), showing the experimental setups for the performance testing of the developed differential system under various environmental conditions including heat, cold, humidity and UV (now Figs S4 – S7)

Reviewer #2 (Remarks to the Author):

1. The suggestion that THz sensors are expensive may be overstating the case, certainly MMICs are available nowadays from commercial foundries operating to 0.16 THz. The authors may want to check out Luna's Terametrix offering? The response seems very sensitive to the SRR height above the sample. Could you comment on how this affects accuracy? Could you use higher order modes too?

Authors' Response:

Thank you for the comment. The main of the presented work is to explore the capabilities of SRR-based microwave NDI systems to detect the erosive wear of coatings in real-time and remotely. Based on the intended operational environment, the system was designed and tested in various environmental conditions with an addition of an AI-based prediction system to it. The presented table was to introduce the readers to the various NDI systems/methods and their applicability to wear detection, rather than a comparison with microwave-based system. Hence, the table was moved to Supplementary Information document to maintain the focus of the paper on the presented works and the observed results.

As mentioned, the accuracy of a system is a critical factor in its performance. It is generally accepted that as the sensitivity of a system increases, there is a trade-off with accuracy. However, in the case of the developed system, we focused on evaluating the resolution of the system in detecting and quantifying variations in the thickness of coatings. By recording and calculating these variations, we aimed to assess the system's ability to provide precise measurements for the wear depth. This information has been added to the Results section of the updated manuscript.

Additionally, the system uses the first dominant mode resonance since it has been extensively studied in the past. In the future, we could go with higher order modes, but that is currently out of the scope of this work.

The following modifications are added to the updated manuscript:

The limitation of THz sensor is updated to just "Cannot be used in moist environments". This comparison table is added to Supplementary Information to keep the manuscript precise and clear, focusing on the presented work and the obtained results.

Results: Erosive Wear Detection of WRC

"The sensitivity and resolution of the developed NDI system were determined from Fig. 2b by analyzing the rate of increase of the operational parameters as the wear depth increased. At a wear depth of 1500 μm , the SRR demonstrated a sensitivity of 4 MHz for every 30 μm of erosive wear (averaged). However, at a wear depth of 2000 μm , the sensitivity increased to about 22 MHz for every 30 μm of erosive wear. The

increase in the SRR's sensitivity at greater wear depths was caused by the higher electromagnetic field concentration interacting with the coating closer to the SRR (**Fig. S2**). In addition to the instantaneous values, the dynamic variations of the system's operational parameters also facilitated the estimation and verification of the wear depth. It can be observed from Fig. 2b that the erosive wear of the WRC-coated SRR-based NDI system resulted in an increase in both the resonant frequency and the Q-factor values at an increasing rate."

2. Is the VNA calibrated? If so state how and if not why not - how do you guarantee repeatability?

Authors' Response:

Thank you for the question. The VNA was calibrated using a Keysight Cal Kit 85521A in the beginning of experimentation. This calibration procedure established a reference for accurate measurements, ensuring that the obtained results through the VNA remained repeatable throughout the experimentation.

The following statements are added to the updated manuscript:

Method: Experimental Setup for Erosion Wear Testing

"The VNA was calibrated using a Keysight Cal Kit 85521A calibration tool which ensured the repeatability in the system's response. The Intermediate Frequency (IF) bandwidth of the VNA was set to 500 Hz to enhance the signal-to-noise ratio of the measurements, with 2001 number of sweeping points and a power level of -5 dB."

3. The Q measured will also depend upon the conductor backing of the dielectric sample.

Authors' Response:

Thank you for your comment. In the case of dielectric coatings, there is a minimal change in the system's response (Q) due to erosive wear of the coating layer. However, in the case of conductor-based coatings, as you pointed out, the Q of the system will be affected by the proximity of the coating to the system. Similarly, the presence of the conductor backing (ground plane) on the dielectric substrate over which the SRR is fabricated, too would have an impact on the Q. For a given geometry of the SRR, the presence of electric and magnetic fields in the substrate would be disturbed by the addition of a conductor backing which could also be a function of the conductor's proximity to the SRR. The equation for the Q of the SRR: $Q = \text{Power}_{\text{stored}}/\text{Power}_{\text{loss}}$ establishes the relation between the proximity of the conductor backing to the SRR with its Q in a variable manner dependent on the geometry of the SRR design. However, the geometry of the system can be modified to compensate for such variations in the Q caused due to the conductor backing of the dielectric sample.

4. While the reference SRR conductor shield is presumably many skin depths thick, preventing field leakage this shield too has the potential to act as a patch antenna - did the authors consider this? Induced currents in the shield will radiate around the shield edges.

Authors' Response:

Thank you for the comment. The copper shielding as seen in Fig. 7a (previous version) indeed has the possibility of acting as a patch antenna, since it was not connected to the ground. The issue was mitigated in the updated design as seen in Fig. 10 (previous version) / Fig. 4 (updated version) in which the center copper shielding was connected with the remaining two shields and was grounded. The subset of the illustration is added below as Fig. R2.

Fig R2. Differential SRR NDI system coated with multi-layer WRC coatings.

The following statements are added to the updated manuscript:

Results: AI-enabled Multilayer WRC Erosion Detection and Estimation

“To achieve this, the planar copper shield in the system design was extended from Port 1 to Port 2, providing complete shielding to both the reference SRR and the MTL (**Fig. 4a**), thus preventing accidental loading of these elements. This design modification also mitigated the potential risk of the copper shielding (over the reference SRR) acting as a patch antenna.”

5. Some very well presented experimental work on environmental effects though.

Authors' Response:

Thank you.

6. Line 305: electric field not Electric.

Authors' Response:

Fixed.

7. The novelty of this approach appears to be in part the application where the only parameter being changed is the dielectric thickness indicating wear. Environmental changes are neatly compensated for through the reference resonator. Further the VNA is replaced by an analogue setup monitoring the resonant frequency of the SRR making for a compact sensor for use on helicopter blades. The significance of this work is relatively minor as a novel NDT technique but may have its own niche application.

Authors' Response:

Thank you for the comment. The need for a coating health monitoring system is prevalent in applications including helicopters, wind turbines, pipelines and so on. Through the implementation of a smart readout circuitry with the developed SRR-based system, the portability, cost-effectiveness, and the remote wireless communication capabilities of the developed wear detection system were enhanced. These system features in addition to the differential mode sensing (that compensated for the environmental noises) would make the developed system of immediate interest in many industries including aviation, marine and oil and gas. Accordingly, we maintain that there is sufficient novelty within our developed system as a novel NDT technique.

The following modifications are added to the updated manuscript:

In the introduction section, the importance, and the potential application of such a coating status monitoring system have been further clarified.

Reviewer #3 (Remarks to the Author):

1. In the last paragraph of the Introduction, the authors indicated a novel differential sensor is introduced. However, what is novel about this differential sensor is not clear compared with previously published microwave differential sensors. Similarly, non-differential sensors have been published previously. So what is significant for the proposed technique? It would be good to expand this paragraph and give further convincing details.

Authors' Response:

Thank you for the comment. The primary and significant contributions of the developed wear detection system include:

- Contactless and Non-Destructive Inspection of WRC erosion using an SRR-based microwave NDI system that can estimate the wear depth based on the variations in the resonant parameters of the SRR in real-time.
- Compensation of the impact of environmental noises on the system's response through the incorporation of an electromagnetically shielded reference SRR element, thereby improving its robustness and usage in harsh environments.
- AI-enabled detection of erosive of multi-layered heterogeneous coatings using the developed differential SRR-based system, with the capability to locate the layer of erosion, thus demonstrating its potential for real-world applications involving multi-layered coatings.
- Implementation of a smart readout circuitry to promote the portability, cost-effectiveness, and the remote wireless communication capabilities of the developed wear detection system.

In comparison to the previous works, the real-time and remote sensing of WRC erosive wear are the two major challenges that are overcome using the proposed system. First, all the measurements recorded using the SRR-based NDI system demonstrate real-time variations in its response with changes in the wear depth of the WRC layer. Second, the portability, cost-effectiveness and wireless monitoring of the system response are enhanced through the integration of a smart monitoring system connected via Bluetooth to a remote mobile device.

The following paragraph is added to the Introduction section to clarify the contribution of the work:

“However, the limited adoption of microwave-based NDI systems for coating wear detection is mainly due to the requirement of specialized equipment and expertise, resulting in high operational costs for reliable measurements. Additionally, erosive wear is an intricate and complex process, often occurring in harsh environments, making it challenging to develop an effective microwave NDI system that can accurately assess wear conditions. The utilization of these systems in eroding environments is impeded by their susceptibility to interference, signal degradation, and the complexity of calibration procedures. Further, the various WRC materials can exhibit different electromagnetic properties, necessitating the development of a system that is adaptable to many different WRCs. In this work, we develop a differential SRR-based NDI system for the real-time detection and monitoring of erosive wear of multi-layer coated surfaces. The WRC-coated microwave NDI system functions by monitoring the variations in the resonant parameters of the embedded SRRs as a function of wear depth of WRCs. Contrary to prior works, the system can detect and locate the eroding layer in multi-layered coatings in addition to detecting the total wear depth. The system consists of a smart wireless readout circuitry for the remote and autonomous monitoring of WRC erosive wear, circumventing the need for a Vector Network Analyzer (VNA). The robustness of the developed system is also verified by observing the system's response in different harsh environments, including various temperature, humidity, and ultraviolet light (UV) exposure conditions.”

2. What is the limitation of WRC total thickness in eqn 5 and 6 (for example, when the effective permittivity does not change)?

Authors' Response:

Thank you for the question. Eq. 5 and 6 establish the relationship between the thickness of the WRC (multi-layered and heterogeneous) and their impact on the effective permittivity of the media. It must be noted that Eq. 5 and 6 are extended versions of Eq. 3 and 4 (for single layer coatings). Hence, any changes in thickness of WRC will in fact affect the effective permittivity of the media, thereby altering the resonant frequency of the SRR as per Eq. 1.

However, during the erosive wear of the 2.5 mm thick epoxy-based coating, the developed system demonstrated a sensing range of 2.2 mm, beyond which the material-electromagnetic interaction was not significant enough to produce an observable change in the system's response. It must be noted that the sensing range of the system is material dependent and would vary for different materials, but for epoxy WRC the total thickness limitation was found to be 2.2 mm. Hence, the system must be calibrated for the selected WRC to obtain the material-specific system performance metrics.

The following statements are added to the updated manuscript:

Results: Theory

“For both single- and multi-layer coatings, erosive wear reduces ϵ_{top} since the electromagnetic field lines interact with the air media more. This results in a decrease in ϵ_{eff} , increasing the resonant frequency observed by the developed system (Fig. 1d).”

Results: Erosive Wear Detection of WRC

“The 50 μm wear depth facilitated the determination of the NDI system's operational range, observed to be ~ 2.2 mm in coating thickness for the epoxy-based WRC investigated. Furthermore, at a thickness of 2.2 mm, 50 μm of erosive wear of the WRC resulted in a resonant frequency increase of 1.4 MHz, due to the decrease in the effective permittivity in the medium, as per Eq. 1 and 2. For the selected coating, at greater thicknesses, the material-electromagnetic interaction was not significant enough to produce an observable change in the system's response.”

“However, note that both the trendlines and the system performance metrics (sensitivity, resolution, and operational range) will depend on the specific WRC being monitored. Hence, proper calibration of the SRR-based system to the specific WRC prior to its installation and usage is necessary for the predictive analysis of the system's response.”

3. The theory in Section 2.1 was clear and easy to follow in terms of detecting multiple layers and defining their thickness and relative permittivities. However, in section 2.2 it is unclear what the permittivity of coated material and the erosive material was. Was the 2.5 mm coated layer verified with measurement? What about the erosive one? This section needs substantial changes to clearly explain the steps for verifying and detecting multiple layers and erosive materials.

Authors' Response:

Thank you for your comment and apologies for the confusion caused during the explanation of the erosive wear tests. The relative permittivity of the epoxy-based coating mentioned in section 2.2 (previous version) is about 2.7 at 2.5 GHz. The permittivity was verified through datasheets and simulations using Ansys HFSS. The thickness of the epoxy coating is ~ 2.5 mm, as verified using the Olympus LEXT OLS5000 3D scanning microscope (see height maps in the updated Figure 2). The erosive wear of the coating layer was performed by rectilinear erosive motion of a mechanical filer over the WRC layer. After each cycle of erosion the filer is removed, so there is no separate erosive material or layer. The updated manuscript incorporates these changes to add more clarity.

The following modifications are added to the updated manuscript:

Method: Experimental Setup for Erosion Wear Testing

“In the case of homogeneous WRC erosion testing, the SRR on the top substrate was coated with 2.5 mm thick industry-grade Bisphenol A and Epichlorohydrin epoxy resin (permittivity of 2.7 at 2.5 GHz, obtained using Ansys HFSS simulations). The resin was cured for 48 hours at standard room temperature and pressure. The process of accelerated erosive wear of the WRC coated system was performed by rectilinear erosive motion of a mechanical filer over the WRC layer. The variations in the resonant parameters of the system’s response, as the WRC eroded, were observed in real-time using the S5065 2-port VNA. At the end of each erosive wear cycle, in order to measure the wear depth and calibrate the system’s response, height maps were obtained using Olympus LEXT OLS5000 3D scanning microscope. In the height map scans, specifically along the scan zone as depicted in **Fig. 2a**, the eroded regions exhibited a high degree of surface roughness due to the physical wear process. Thus, the heights were averaged in the eroded regions to determine the average wear depth. The erosive wear experiment was carried out for 17 cycles until the epoxy layer was completely eroded away. Similarly, in the case of erosive wear of heterogeneous WRCs, the system was coated with three layers of WRCs, representing a multi-layer coating. The selected WRCs included multiple sheets of polyethylene coat (3 sheets of 200 μm thickness each), polyimide coat (12 sheets of 50 μm thickness each) and polyurethane protection film (4 sheets of 150 μm thickness each). The wear testing process involved the meticulous stripping of each WRC sheet while recording the system's response in real-time using a VNA.”

4. I assume erosion depth in figure 6 c and d are produced using eqn 8 considering fr and Q factor? Need to make this point clear.

Authors’ Response:

Thank you for the comment. Indeed, the trendlines illustrated in Fig. 6c and 6d (previous version) are produced using Eq. 8. We have improved the clarity of the explanation in the updated version.

The following paragraph was updated in the modified version of the manuscript in the section Results: Erosive Wear Detection of WRC

“In the obtained results, the relationship between the wear depth and the system’s response is modeled using a Gaussian Green function that can precisely capture the decay of the electromagnetic field (from the SRR) in free space⁴⁸⁻⁵¹, given by:

$$d = a_1 * e^{-\left(\frac{x-b_1}{c_1}\right)^2} + a_2 * e^{-\left(\frac{x-b_2}{c_2}\right)^2} \quad (8)$$

where d is the depth of erosion, x is the system operational parameter and a , b , and c refer to the amplitude, median, and standard deviation of the function, respectively. For the resonant frequency increase, the terms in Eq. 8 were $a_1=2.861e+13$, $a_2=2.171$, $b_1=19.04$, $b_2=10.1$, $c_1=2.929$, $c_2=34.47$. For the Q-factor increase, the terms were $a_1=2.335e+21$, $a_2=1.802e+07$, $b_1=9.141$, $b_2=120.6$, $c_1=0.9966$, $c_2=33.23$ (Fig. 2b). The fitted trendlines resulted in R-squared values of 0.96 and 0.99, for resonant frequency and Q-factor, respectively. Following the calibration phase, the system is deployed for usage, during which it continuously captures the response of the coating and applies it to the pre-established trend lines. By comparing the real-time response of the system to the trend lines, the system can determine and estimate the wear depth of the coating as erosion occurs.”

5. Section 3: it was mentioned, "Above it was demonstrated that microwave SRR sensors enable the real-time detection", however it was not real-time but post-processing.

Authors’ Response:

Thank you for your comment. We would like to emphasize that the measurements obtained using the SRR-based NDI system demonstrate real-time variations in the system's response as the wear depth of the WRC layer changes. The erosion was detected continuously in real-time during the entire erosion process and was not post-processed. However, to determine the depth of the erosion the correlated to the response of

our system, the erosion was paused at 17 arbitrary time points (denoted “erosion cycles” in the manuscript) and the WRC-coated sensor was taken to an optical profilometer to measure the erosion depth. We’ve clarified this point better in the updated manuscript:

The following updates are made in the modified version of the manuscript:

Methods: Experimental Setup for Erosion Wear Testing

“The variations in the resonant parameters of the system’s response, as the WRC eroded, were observed in real-time using the S5065 2-port VNA. While the erosion was monitored in real-time, after 17 different amounts of erosion, each denoted as an erosion cycle, the WRC-coated sensor was disconnected from the VNA to determine the depth of erosion and calibrate the system’s response. Height maps were then obtained using Olympus LEXT OLS5000 3D scanning microscope.”

Results: Erosive Wear Detection of WRC

“Thus, the instantaneous values of the resonant frequency and the Q-factor of the SRR (post calibration of the system to the specific WRC), enabled the real-time estimation of the wear depth.”

- Section 3.1: I suggest including the equivalent circuit for the differential sensor and producing simulation results for the circuit with verified EM simulation. Interesting to know particularly the effect of shielding plates.

Authors’ Response:

Thank you for the comment. The main objective of conducting the equivalent circuit analysis is to gain a better understanding of the system's operation and response to the erosive wear of WRC coatings. However, to maintain precision and focus on the manuscript, the detailed design and operation of the system took precedence. As a result, the presentation of the equivalent circuit model has been moved to the Supplementary Information section.

In the differential system design presented in Section 3.1 (previous version), the introduction of the copper shields would introduce two capacitances, C_{ms} and C_{srs} , which would be parallel to the capacitances $C_t = C_{srr} + C_c + C_{srg} + C_{mg}$. Here C_{srr} is the intrinsic capacitance in the SRR, C_c is the coupling capacitance between the SRR and the MTL, and C_{mg} and C_{srg} are the capacitances formed between the MTL and ground plane and SRR and the ground plane, respectively. This ultimately reduces the resonant frequency, which was experimentally observed to be in the range of tens of MHz). However, when the system is coated with a WRC, the total capacitance increases by the value C_{wrc} (a function of the relative permittivity and thickness of WRC).

Fig. R3 Distribution of capacitances in the developed system (c) without an electromagnetically shielded reference resonator, and (d) with an electromagnetically shielded reference resonator

The following modifications are added to the updated manuscript:

The equivalent circuit model and its explanation have been added to the Supplementary Information.

“The equivalent lumped circuit model analysis was performed for the developed system design for better understanding of the operation and response of the system to the erosive wear of the WRC coatings. The

circuit model was designed and validated through simulations using Advanced Design System. The operation of the SRR was comparable to that of an LC resonator circuit with $f_r = 1/(2\pi\sqrt{LC_t})$, where L is the inductance due to the current loop path on the SRR, and C_t is the total capacitance in the SRR ring. Variations in the C_t value (a function of effective permittivity) led to changes in the resonant frequency while the inductance L was kept constant. When the system was not coated, the total capacitance (C_t) consisted of three parallel capacitances resulting in $C_t = C_{srr} + C_c + C_{srg} + C_{mg}$ (**Fig S3c**). Here, C_{srr} is the intrinsic capacitance in the SRR, C_c is the coupling capacitance between the SRR and the MTL, and C_{mg} and C_{srg} are the capacitances formed between the ground plane and the MTL and SRR, respectively. However, when the system was coated with a WRC, the total capacitance increased by the value C_{wrc} (a function of the relative permittivity and thickness of the WRC) and thus, the resonant frequency varied as per Eq. S1.

$$f_r = \frac{1}{2\pi\sqrt{L(C_{srr}+C_c+C_{srg}+C_{mg}+C_{wrc})}} \quad (S1)$$

Additionally, in the system model presented in Fig. 3 (main manuscript), the introduction of the copper shields over the reference SRR would introduce additional capacitances C_{ms} and C_{srs} , which are the capacitance formed between the copper shield and the MTL and reference SRR, respectively. These capacitances invariably increase the total capacitance, ultimately reducing the resonant frequency.”

7. The relation between the design theory of multilayer detection and the results presented in Section 3.2 is unclear (how this concept has been used to detect environmental changes). Or is it simply a single-layer measurement with changes in the dimension and permittivity of the single layer?

Authors’ Response:

Thank you for your comment. We apologize for the confusion that has arisen due to the naming of Section 3 as, “Multilayer WRC Erosive Wear Sensing by a Differential Microwave Sensor” (previous version).

The following modifications are added to the updated manuscript:

In the updated manuscript, the flow of the work is as follows:

- The first two parts of the Results section begin with the operational principle and the experimental results of the SRR-based system for detection of erosive wear of single layer coatings (in this case the epoxy resin)
- In the third part of the Results section, the developed system is upgraded to a differential SRR-based model to compensate for the environmental noises.
- Following that, in the fourth part of the Results section, the differential system is tested to detect erosive wear of multi-layer coatings and additionally incorporates an RNN-based model to predict wear depth through predictive analysis methods.
- Lastly, the SRR-based system is integrated with a smart monitoring readout circuitry which makes the system portable and cost-effective.

8. It would be good to present a table comparing the results from this work with state-of-the-art.

Authors’ Response:

Thank you for the comment. Indeed, prior research efforts have been carried out in the past for coating health monitoring using microwave-based resonators. However, the primary focus of these past efforts has involved the detection of cracks, corrosion, and defect formation, and not erosive wear. Closely related works include detection of coating health status that demonstrate either partial or full coating detachment. These related works are cited in the introduction section of the updated manuscript. However, in order to avoid system benchmarking, no direct quantitative comparison was made since the application of the system in each case was different from each other (including the materials used).

REVIEWERS' COMMENTS

Reviewer #1 (Remarks to the Author):

The authors have response all of the comments and they are satisfactory.

Reviewer #2 (Remarks to the Author):

The updates to the work presented are satisfactory and improve the readability of the paper. Whilst I am not wholly convinced of how this would be deployed into an industrial setting I am convinced the work is of merit and a useful addition to the literature. The revised paper detailing measurements taken and calibrations deployed would allow for the work to be replicated by other researchers.

Reviewer #3 (Remarks to the Author):

Thank you for your response and no further question from me.